# Neither Incretin or Amino Acid Responses, nor Casein Content, Account for the Equal Insulin Response Following Iso-Lactose Loads of Natural Human and Cow Milk in Healthy Young Adults

**DOI:** 10.3390/nu14081624

**Published:** 2022-04-13

**Authors:** Paolo Tessari, Alessandro Toffolon, Monica Vettore, Elisabetta Iori, Anna Lante, Emiliano Feller, Elisabetta Alma Rocco, Monica Vedovato, Giovanna Verlato, Massimo Bellettato

**Affiliations:** 1Department of Medicine (DIMED), Diabetes and Metabolism Division, University of Padova, 35128 Padova, Italy; ale.toffolon@gmail.com (A.T.); vettore.monica@unipd.it (M.V.); elisabetta.iori@unipd.it (E.I.); elisabettaalmarocco.er@gmail.com (E.A.R.); monica.vedovato@aopd.veneto.it (M.V.); 2Department of Agronomy, Food, Natural Resources, Animals & Environment (DAFNAE), University of Padova, 35123 Padova, Italy; anna.lante@unipd.it; 3Centrale del Latte di Vicenza Spa, via A. Faedo 60, 36100 Vicenza, Italy; feller@centralelatte.vicenza.it; 4Department of Pediatrics, Padova City Hospital, via Giustiniani 1, 35128 Padova, Italy; verlatogiovanna@gmail.com; 5Department of Pediatrics, Vicenza City Hospital, viale Rodolfi, 37, 36100 Vicenza, Italy; massimo.bellettato@ulssvicenza.it

**Keywords:** amino acids, casein, cow milk, dietary supplements, human milk, incretin, insulin, whey proteins

## Abstract

Human milk contains <50% less protein (casein) than cow milk, but is equally effective in insulin secretion despite lower postingestion hyperaminoacidemia. Such potency of human milk might be modulated either by incretins (glucagon-like polypeptide-1,GLP-1); glucose-inhibitory-polypeptide, GIP), and/or by milk casein content. Healthy volunteers of both sexes were fed iso-lactose loads of two low-protein milks, i.e., human [Hum] (*n* = 8) and casein-deprived cow milk (Cow [↓Cas]) (*n* = 10), as well as loads of two high-protein milks, i.e., cow (*n* = 7), and casein-added human-milk (Hum [↑Cas]) (*n* = 7). Plasma glucose, insulin, C-peptide, incretins and amino acid concentrations were measured for 240′. All milks induced the same transient hyperglycemia. The early [20′–30′] insulin and C-peptide responses were comparable among all milk types apart from the low-protein (Cow [↓Cas]) milk, where they were reduced by <50% (*p* < 0.05 vs. others). When comparing the two high-protein milks, GLP-1 and GIP [5’–20’] responses with the (Hum [↑Cas]) milk were lower (by ≈2–3 fold, *p* < 0.007 and *p* < 0.03 respectively) than those with cow milk, whereas incretin secretion was substantially similar. Plasma amino acid increments largely reflected the milk protein content. Thus, neither casein milk content, nor incretin or amino acid concentrations, can account for the specific potency of human milk on insulin secretion, which remains as yet unresolved.

## 1. Introduction

Milk is a complex and complete nutritional substrate [1] endowed with many functional properties, largely due to a variety of substances and hormones, as well as of (oligo)peptides generated from the gut digestion of proteins [2,3,4,5]. Milk composition is species specific [1,6,7]. Compared to cow milk, human milk contains more lactose (≈5 vs. ≈7 g/100 mL, respectively), nearly the same amount of lipids and calories, but <50% less protein, depending on the lactation period [6,7,8,9]. Moreover, the protein fraction(s) are qualitatively and quantitatively different: the proteins are mainly in the soluble whey-fraction in human milk, whereas the (insoluble) casein(s) are predominant in cow milk [6,7,8,9]. Human milk is also rich in many hormones that regulate glucose homeostasis [10], and exhibits anti-inflammatory and antioxidant activities [11]. Although an infant’s growth is physiologically sustained by human milk, the latter can be replaced, and/or completed by, cow-derived formula-milk products [12,13] containing a reduced amount of protein (casein) as well as of salts. Indeed, an excessive intake of protein is not recommended in infants [14,15,16,17], because of the risks of abnormal weight gain in early childhood, and development of metabolic syndrome-related abnormalities later in life [15,16,17]. Mechanistically, these undesired effects were associated to greater insulin, insulin-like growth factor-1 (IGF-I) and amino acid concentrations following high-protein, cow-derived milk products, than those following human milk [17]. As a matter of fact, the role of anabolic hormones is crucial in orchestrating, along with the amino acids, infant growth.

Recently, Gunnerud et al. [18] first reported that the iso-lactose loads of cow and human milk stimulated insulin secretion to the same extent, despite a much lower protein concentration, and a less pronounced hyperaminoacidemia following human than cow milk. This finding appeared somehow unexpected, because amino acids are also potent stimulators of insulin secretion, in addition to, and synergistically with, glucose-derived lactose [19,20,21,22]. Therefore, a lower insulin response would have been expected following human milk.

As possible explanations for such a mismatch, at least two hypotheses could be forwarded: (1) Could the reported “equal” potency between human and cow milk on insulin stimulation be associated to a greater incretin response with the former, thus offsetting the (opposite) effects due to the lower protein content and the less marked postingestion hyperaminoacidemia? Incretins, among them the glucose-inhibitory-polypeptide (GIP) and the glucagon-like polypeptide-1 (GLP-1), are gastrointestinal hormones that are stimulated by nutrient ingestion and directly enhance insulin secretion [23]. Unfortunately, in the referenced study [18], the incretin response was incompletely reported, and the relatively short study duration (2 h) could have prevented the detection of possible late-occurring hormonal and metabolic changes. Alternatively: (2) Could the much lower casein content of human than cow milk have minimized any possible interference conveyed by casein itself on the whey protein-mediated insulin stimulation of human milk? Since casein coagulates in the acid pH of the stomach [24], the so-formed curd could impair whey protein intestinal digestion and amino acid absorption and, ultimately, compromise substrate-mediated signaling on insulin secretion.

Therefore, in this study we addressed the above hypotheses by testing, in young healthy volunteers, the effects of cow and human milk, as well as the manipulation of their casein content, on glucose, insulin, C-peptide, GLP-1, GIP and plasma amino acid concentrations. The insulin secretory capacity was tested using both ”natural” and experimentally-modified cow and human milk, the former prepared by decreasing casein concentration down to that of human milk, the latter by increasing it up to the cow-milk value.

## 2. Materials and Methods

### 2.1. Participants

Sixteen young healthy volunteers (7 males, 9 females; age: 23.1 ± 0.6 yrs.; BMI: 22.1 ± 0.8 kg/m^2^) were enrolled. The study was approved by the Ethical Committee of the Padova University and City Hospital (N° 2861P, on 8 July 2013, with a followed-on amendment in April, 2021) and was performed according to the guidelines of the 2013 Helsinki Declaration [25]. The protocol is registered at the ClinicalTrial.gov.site (ID: NCT04698889) (accessed on 4 February 2022). The studies were effectively started on April 2015, intermittently halted for lack of volunteers and/or support personnel, and finally completed (including all the analytical measurements and calculations) by 30 June 2021. The volunteers were sequentially recruited as they became available for the studies. Although the investigators’ intention-to-treat scheme was to apply a random allocation of the same number of volunteers to all the four arms of the protocol, some participants unpredictably refused to participate in some of the planned studies. Therefore, neither the number of participants of each test type, nor subject’s participation in all the planned tests, could be observed (See Appendix A, for the study planning and recruitment).

### 2.2. Type and Composition of Tested Milks

Four types of iso-lactose milk loads were administered in separate occasions: (1) natural cow whole milk (designated as: [Cow]) (to seven participants, 4 F, 3 M, BMI: 21.3 ± 1.3 Kg/m^2^); (2) natural human milk (designated as: [Hum]) (to eight participants, 3 F, 5 M, BMI: 23.8 ± 1.0 Kg/m^2^); (3) a cow-based whole milk, with a decreased total protein concentration (<1 g/100 mL) obtained by reducing its natural casein content, designated as: Cow [↓Cas], and with a target whey protein-to-casein ratio approaching that of human milk (between 60/40 and 70/30) on the first ≈1–6 months of lactation [9] (to ten participants, 5 F, 5 M, BMI: 22.4 ± 1.1 Kg/m^2^); and, finally, (4) a casein-added human milk (designated as: Hum [↑Cas]), matching the casein as well as the total protein content of natural cow milk, but with a casein-to-whey protein ratio (≈80/20) approaching that of cow whole milk [6] (to seven participants, 3 F, 4 M, BMI: 22.3 ± 1.1 Kg/m^2^). Therefore, a total of 32 test meal applications were performed. Six participants participated in three different studies, four in two different studies, whereas six participants were studied only once. Of the 32 studies, male participants contributed to a total of 17 studies, female participants to the remaining 15, both sexes being thus represented in a fairly well-balanced proportion in each experimental group. In repeated different tests performed on the same subject, at least three weeks were spaced between each of them. The analytical composition of the four milk types is reported in Table 1, whereas additional details about their preparation are reported in Appendix A.

### 2.3. Milk Provision and Preparation

#### 2.3.1. Cow Milk

The cow milk was a commercial, pasteurized whole cow milk (Parmalat, Collecchio, Parma, Italy], with a label-reported concentration of 3.4% (g/vol) protein, 4.9% carbohydrate (lactose) and 3.5% fat. To this milk, lactose was added with the aim to raise its concentration to approximately 7 g/100 mL (Table 1). The milk was administered to each subject in volumes calculated to provide lactose at ≈0.35–0.36 g/kg body weight (corresponding to ≈25 g lactose in a reference 70 kg subject) (Table 1).

#### 2.3.2. Human Milk

Human milk was provided by the “Bank of Donated Human Milk” of the Pediatric Dept. of the San Bortolo City Hospital (Vicenza, Italy), following a carefully-controlled procedure [27] approved by the Italian Ministry of Health (According to the guidelines of the Italian Ministry of Health (issued 5 December 2013). Published in the G.U. Serie Generale, n. 32, 8 February 2014). Milk batches were temporarily kept by the donating women in their home freezers, then collected, processed, stored and utilized following the protocol guidelines (See Appendix A for details). The milk batches utilized in the study were a mixture of milk samples collected by different donating women over the first ≈6 months of lactation. The volumes of administered human milk were calculated to provide the target lactose load [≈0.35–0.36 g/kg BW], to each study participants, by assuming a theoretical lactose concentration in human milk of ≈7 g/100 mL. As a matter of fact, the composition of human milk could not be directly measured before the studies, because we were committed not to thaw the milk batches days before the in vivo administration, for safety reasons. Therefore, such a time-restriction prevented a timely, pretest analysis by the laboratory. However, the effectively-administered lactose loads were re-calculated after the test following direct measurements of lactose concentrations in milk samples (Table 1) [See also Appendix A].

#### 2.3.3. Low Casein Cow-Milk

The Cow [↓Cas] milk was an experimentally prepared, low-protein cow milk, with a ≈1% (w/vol) total protein concentration, but with a target WP/casein ratio approaching that of human milk in early-to-middle lactation (between 60/40 and 70/30) [9]. This milk was prepared by diluting a predetermined volume of cow whole milk with distilled water, in order to decrease the natural casein concentration to the desired value (i.e., that of natural human milk), followed by additions of calibrated amounts of (cow-derived) whey-proteins (Table 1) (See Appendix A for further details).

Lactose was added to this milk too, with the aim to raise its total concentration (resulting from the sum of the added and the natural lactose content of cow milk), to approximately 7 g/100 mL (Table 1). The administered milk volume was again calibrated to provide lactose in the individualized amount of ≈0.35–0.36 g/kg body weight (i.e., ≈25 g lactose in a reference 70 kg subject) (Table 1). Furthermore, fat (as milk cream) and salts (sodium, potassium, iron, calcium, phosphorus and magnesium, obtained from available hospital solutions) were also added to this milk, in amounts titrated to approximately match their (theoretical) concentrations to those of natural cow milk [28,29] (See Appendix A for additional, detailed specifications).

#### 2.3.4. Casein-Added Human Milk

The Hum(↑Cas) milk] was prepared by adding cow casein (as calcium caseinate, Milk Protein 90, +Watt, Ponte San Nicolò, Padova, Italy), to natural human milk, in amounts calculated to attain a target total protein concentration similar to that of natural cow milk, but a WP-to-casein ratio similar to that of cow milk (≈20/80) [6]. Furthermore, in this milk type, the milk volumes to be administered were calculated to provide ≈0.35–0.36/kg BW lactose in each subject, under the assumption of an average lactose concentration in human milk of ≈7 g/100 mL (Table 1). The actual lactose loads administered to each subject with this milk too, were directly measured and recalculated after the studies (See Appendix A).

### 2.4. Experimental Procedures

The volunteers were admitted to the clinical study unit at ≈08:00 a.m. after >12 h. of overnight fast, and placed in a dedicated armchair. A 20 g cannula was inserted in an antecubital vein for blood withdrawal. After two baseline samples spaced by ≈10′, the milk load was administered and entirely consumed within 2′–5′. Starting from the end of milk ingestion (t = 0′), blood samples were collected at min 5′, 10′, 20′, 30′, 60′, 90′, 120′, 180′, and 240′, then immediately transferred to two series of plastic tubes, containing either Na-EDTA (6% w/vol), for glucose, amino acid, insulin and C-peptide determinations, or a protease inhibitor (EMD Millipore Corporation, Merck KGaA, Darmstadt, Germany), for GIP and GLP-1 assay. The tubes were kept on ice and delivered to the laboratory every 60′–90′. After centrifugation, aliquots of plasma were immediately frozen and kept at −80 °C until assayed. The participants rested in the armchair and did not eat anything over the entire duration of the test. Only drinking water was allowed.

### 2.5. Biochemical Analyses

The primary outcomes of the study were to measure plasma glucose, insulin, C-peptide and incretin concentration over the test. The secondary outcome was to measure plasma amino acids.

Plasma glucose concentration was determined by the glucose oxidase method, using a Yellow Spring glucose analyzer (Yellow Spring Inc., Yellow Springs, OH, USA). Insulin, C-peptide, GIP and GLP-1 plasma concentrations were determined by ELISA (Merck-Millipore Corporation, Merck KGaA, Darmstadt, Germany). The intra- and interassay variability (expressed as coefficient of variation, in %) were 2.2 and 4.5, respectively, for insulin, 6.12 and 9.21 for C-peptide, 7.4 and 8 for GLP-1, and 6.5 and 3.4 for GIP. The GLP-1 data of one subject out of the total ten of the Cow [↓Cas] milk group, as well as those of one subject out of eight participants of the Hum milk group, were excluded from calculation because they were considered outliers (See Appendix A). In one subject of the Hum [↓Cas] milk group, GIP was not determined for lack of plasma sample. Plasma amino acid concentrations were determined by gas chromatography–mass spectrometry (GCMS) using modifications of published methods [30,31,32] (See Appendix A for additional details and the CVs of plasma amino acid analysis). Lactose, fat and protein concentrations in the administered milks, were analyzed by standard laboratory methods.

### 2.6. Statistical Analysis

The data statistical analysis was performed either on absolute concentrations, or on their relative change (i.e., delta, Δ) in respect to baseline data, depending on the variability (expressed as coefficient of variation, CV) of the parameter’s baseline data among the groups’ means. Absolute concentration data were used where the CV was <2% (i.e., glucose), whereas Δ-changes were used when the CV was >2% (for the other parameters). Such a criterion was arbitrarily adopted to account for intertest (i.e., for intersubject group) variability. The incremental area(s) under the curve (iAUC) within specific time intervals (as detailed), were calculated using the trapezoidal approach, i.e., by calculating each area within two experimental times and adding them up. The one-way analysis of variance (ANOVA) was applied to analyze/compare one single set of data (i.e., fasting plasma glucose, insulin; the 240′ iAUC of any parameter, etc.) among the four experimental groups, followed by post hoc analysis using the Fisher’s LSD test. To analyze/compare multiple measurements of a single parameter over time between the experimental groups, the data were analyzed using the two-way ANOVA for repeated measurements, followed by the Fisher’s LSD test. The two tailed Student’s *t* test for unpaired data was used in some instances for head-to-head comparisons, as specifically indicated. The Statistica Software (version 10, TIBCO Sofware Inc., Palo Alto, CA, USA) was employed. A *p* value of <0.05 was considered statistically significant. Although the original study design was a cross-over one, it shifted to a partially-parallel one (i.e., one with independent groups), in the course of the protocol, because of the unavailability of some participants to undertake the planned study number, as reported above. Therefore, we adjusted power analysis to that applicable to four independent groups. Power analysis, sufficient to detect a ≈30% difference between mean pairs, with a RMSSE (root mean standardized square error) of <0.5, a power of 80%, and a two-sided level of significance of 0.05%, indicated a sample size of at least 29 test meal applications, therefore, below the total number of studies here executed.

## 3. Results

The study flow chart, participants’ enrollment and drop-outs before test initiation, their effective participation in the planned different milk tests, and other details, are reported in Appendix A. Each participant succeeded in completing the initiated test(s) for the planned 4 h duration. No participant reported any relevant side-effect except for minor abdominal discomfort in three of them, independent of the test type.

### 3.1. Test Milk Composition and Loads

The milks’ composition is visually depicted in Figure 1, and analytically reported in Table 1. The post hoc measured lactose concentrations in the administered milks were quite similar among the four milk types (Table 1). The actual lactose loads (in g/kg BW) resulted in being 0.347 with [Cow] milk, 0.343 with [Hum] milk, 0.337 with the Cow [↓Cas] milk, and 0.336 with the Hum [↑Cas] milk. The administered milk volumes were also quite similar across the four experimental groups, ranging from 4.82 to 5.47 mL/Kg body weight (Table 1). The measured total protein concentration in natural human milk was within the range of published milk data of the first six months of lactation (i.e., ≈1.0–1.4 g/100 mL) [9]. The calculated casein/WP ratio in the Cow [↓Cas] milk was [29/71] (The difference between the post hoc calculated [WP/Cas] ratio [≈70/30] of the Cow [↓Cas] milk (from Table 1), and the target value of Human milk [≈60/40] [9], as post hoc verified in our laboratory (see Appendix A and Figure A1), was due to the impossibility of measuring actual human milk protein composition before the test (see: Methods). Therefore, in the calculations applied to the preparation of the Cow [↓Cas] milk, we provisionally assumed a [WP/Cas] ratio [≈70/30], probably closer to that of the early two weeks of lactation [9]), than in the Hum [↑Cas] [83/17] (Table 1). Thus, casein deprivation and whey protein addition to Cow milk succeeded in the reversal of the casein/WP ratio of natural cow milk, to a value close to that of natural human milk in the early-to-middle lactation period(s) [9]. Conversely, casein addition to human milk succeeded in the reversal of the casein/WP ratio of natural human milk to the value found in natural cow milk [6].

The fat content of cow milk was ≈50% greater than that of natural human milk, ≈60% greater than that of the Cow [↓Cas] milk, and ≈90% greater than that of the Hum [↑Cas] milk (Table 1). A strict statistical comparison of fat content among the four milk types (by one-way ANOVA) was prevented by the assumption of a (label-reported) 3.5% fat content in the commercial cow milk, not directly verified directly in our laboratory and therefore was not affected by analytical variations, as opposed to direct measurements of fat in the other milk types.

Glucose. Postabsorptive plasma glucose concentrations ranged between 90–92 mg/100 mL. Following the administration of each milk type, plasma glucose increased modestly (+10–20%) but significantly (*p* < 0.000001 by two-way ANOVA for repeated measurements, time effect) (Figure 2a), to peak values of 105–112 mg/100 mL, attained after 20′–30′ in all milk tests. No statistical difference in glucose concentrations was observed among the four milk types at any time point, selected intervals or incremental areas under the curve (data not reported) (iAUC) (*p* > 0.5 by ANOVA, either as group or interaction effect).

Insulin. Postabsorptive plasma insulin concentrations ranged between ≈10 and ≈16 μU/mL, without differences between the four studies. Despite the small increments of plasma glucose, plasma insulin increased in all studies by ≈3.5–4 fold vs. baseline (*p* < 0.00001 by two-way ANOVA for repeated measurements, time effect), to peak values at 30′ with all test milks (Figure 2b). The Cow, Hum and Hum [↑Cas] milk determined the same insulin increments (i.e., not different among the groups, *p* > 0.05 at any time point, by two-way ANOVA for repeated measurements, group effect). In contrast, at 10′, 20′ and 30′ the insulin increments with the Cow [↓Cas] milk were between ≈1 and ≈4-fold lower than those with Cow, Hum and Hum [↑Cas] (*p* < 0.05–<0.006, by two-way ANOVA for repeated measurements, group effect, followed by the Fisher’s LSD test) milks (Figure 2b).

### 3.2. Glucose

Postabsorptive plasma glucose concentrations ranged between 90–92 mg/dL. Following the administration of each milk type, plasma glucose increased modestly (+10–20%) but significantly (*p* < 0.001 by ANOVA for repeated measurements, time effect) (Figure 2a), to peak values of 105–112 mg/dL, attained after 20′–30′ in all studies. No statistical difference in glucose concentrations was observed among the four milk types at any time point, selected intervals or incremental areas under the curve (iAUC) (data not reported) (*p* > 0.5 by the two-way ANOVA for repeated measurements, group effect).

### 3.3. Insulin

Postabsorptive plasma insulin concentrations ranged between ≈10 and ≈16 μU/mL, without differences between the four studies. Despite the small increments of plasma glucose, plasma insulin increased in all studies by ≈3.5–4 fold vs. baseline (*p* < 0.0001 by the ANOVA for repeated measurements, time effect), to peak values at 30′ with all test milks (Figure 2b). The Cow, Hum and Hum [↑Cas] milks determined the same insulin increments (i.e., not different among the groups, *p* > 0.05 at any time point, by two-way ANOVA for repeated measurements, group effect). In contrast, at 10′, 20′ and 30′ the insulin increments with the Cow [↓Cas] milk were between ≈1 and ≈4-fold lower than those with Cow, Hum and Hum [↑Cas] (*p* < 0.05; < 0.006, by two-way ANOVA for repeated measurements, group effect, followed by the Fisher’s LSD test) milks (Figure 2b).

### 3.4. C-Peptide

Postabsorptive plasma C-peptide concentrations ranged between 1.0 to 1.2 ng/mL, without differences among the four studies, and increased significantly paralleling the insulin increase (Figure 2c). The C-peptide increments with Cow [↓Cas] milk at 20′ and 30′ were lower those following either the Cow (*p* = 0.011 and *p* = 0.016), Hum (*p* = 0.067 and *p* < 0.005), or Hum [↑Cas] (*p* < 0.01 and *p* < 0.004) milk (Figure 2c).

### 3.5. GLP-1 and GIP

Postabsorptive plasma GLP-1 concentrations ranged between 2.5 to 4.0 pM, without differences between the four studies. Following Cow milk ingestion, GLP-1 concentrations increased to a variable extent over the 4 h study (*p* < 0.00001 by two-way ANOVA for repeated measurements, time effect, and *p* = 0.01, interaction effect). In all groups, the pattern of GLP-1 increments (including, to a less extent, that following Hum [↑Cas] milk), was approximately biphasic, with an early secretion peak at 20′–30′, followed by a second one at 60′–90′ (Figure 3a). With Hum milk, GLP-1 peaked at 30′ as well, but the increment was modest and slowly decreased thereafter. With the Cow [↓Cas] milk, there was a marked and progressive GLP-1 increase, peaking between 60′–90′ and slowly decreasing thereafter. The GLP-1 increments with Cow milk at 10′ were greater than those observed with either Hum (*p* = 0.04) or Hum [↑Cas] milk (*p* < 0.04); greater at 20′ than those with either Hum (*p* < 0.035) or Cow [↓Cas] milk (*p* < 0.025); and greater at 60′ than that with Cow [↓Cas] milk (*p* < 0.045) (by ANOVA and the Fisher’s LSD test). At 90′, the increment with Cow milk tended to be greater, albeit insignificantly (*p* = 0.06) than that with Hum milk.

Due to the biphasic mode of GLP-1 response(s) (Figure 3a), we arbitrarily calculated their iAUCs separately, i.e., both within the [5′–20′] and the [60′–240′] time intervals. With Cow milk, the [5′–20′] GLP-1 iAUC was greater than that with either Hum (by ≈7-fold, *p* < 0.002), Cow [↓Cas] (by ≈2-fold, *p* < 0.035) or Hum [↑Cas] (by ≈5-fold, *p* < 0.007) milk (Figure 4a). In contrast, the [60′–240′] GLP-1 iAUC was not significantly different among the groups, although it was ≈1-fold to ≈5-fold greater with Cow [↓Cas] milk than with Cow (*p* = 0.094), Hum (*p* = 0.056) and Hum [↑Cas] (*p* = 0.087) milk types (Figure 4c).

Postabsorptive GIP concentrations ranged between 1.7 to 1.9 pg/L, without differences among the four studies. Following milk administration, GIP concentrations increased to a variable extent over the 4 h study (*p* < 0.000001 by two-way ANOVA for repeated measurements, time effect). GIP response was biphasic, with the first peak occurring between 10′ and 30′, and the second one between 90′ and 120′ (Figure 3b). The GIP increment with Cow milk at 10′ was greater than that with the Cow [↓Cas] milk (*p* < 0.03) and not statistically different from those of other milk types at any other time point. However, with Cow milk the [5′–20′] GIP iAUC was ≈4-fold greater than that with the Cow [↓Cas] milk (*p* < 0.012), as well as 3-fold greater than that with the Hum [↑Cas] milk (*p* < 0.03) (Figure 4b) (by one-way ANOVA, followed by the Fisher’s LSD test). Within the [60′–240′] interval, the GIP iAUC was not significantly different among the groups (Figure 4d).

### 3.6. Plasma Amino Acids

Total plasma amino acids increased with all milk loads (*p* < 0.0001 by two-way ANOVA for repeated measurements, time effect) (Figure 5a), to an extent grossly reflecting their content in the administered milks, peaking at ≈10′–30′ with Cow, Hum and Cow [↓Cas] milks, and at ≈60′ with the Hum [↑Cas] milk. At 60′, the increment of total amino acids with the Hum [↑Cas] was greater than that following both the Hum (*p* < 0.003) and the Cow [↓Cas] (*p* < 0.009) (by one-way ANOVA and the Fisher’s LSD test). Plasma BCAAs increased, although with variable extent, with all milk types too (*p* < 0.0001 by two-way ANOVA for repeated measurements, time effect) (Figure 5b), peaking at 20′–30′ with Cow, Hum and Cow [↓Cas] milks, and at 90′ with the Hum [↑Cas] milk. At 20′, 60′ and 90, the increments of BCAA concentrations with the Hum [↑Cas] milk were greater than those with other milk types (Figure 5b). The sum phenylalanine (Phe) and tyrosine (Tyr) (Figure 5c) increased, although with a variable extent, also with all milk types (*p* < 0.002 by two-way ANOVA for repeated measurements, interaction effect), again peaking at 20′–30′ with Cow, Hum and Cow [↓Cas] milks, and at 90′ with the Hum [↑Cas] milk. There was no significant difference in the sum of Phe and Tyr increments among the groups at any time point.

The sum of NEAA increased, although to a variable extent, with all milk types too (*p* < 0.002 by 2-WAY ANOVA for repeated measurements, interaction effect), again peaking at 20′–30′ with Cow, Hum and Cow [↓Cas] milks, and at 90′ with the Hum [↑Cas] milk. There was no significant difference in the sum of NEAA increments between the groups at any time point.

The iAUC of the sum of all amino acids, the BCAAs, Phe and Tyr, as well as of the NEAAs, within either the first or the final two hours following milk ingestion, are shown in Figure 6. The [0′–120′] iAUCs were greater than those of the [120′–240′] interval (*p* < 0.001 or less in any amino acid group, The [120′–240′] areas were lower (*p* = 0.001) than [0′–120′] ones (by the paired *t* test within each AA group) in each experimental milk group, except for the that of NEAAs in the Cow milk group, showing a fast amino acid absorption. The iAUCs of the sum of all amino acids, as either the [0′–120′] or the [120′–240′] iAUC (Figure 6a), were not significantly different among the four milk types, possibly because of large intragroup variations. The iAUC of the BCAA with Hum milk was lower than that with Cow milk (*p* < 0.03) in the [120′–240′] interval, as well as lower than that with Hum [↑Cas] in the [0′–120′] interval (*p* < 0.03). Conversely, with Hum [↑Cas] milk the iAUC of the BCAA was greater than that with the Cow [↓Cas] milk in the [0′–120′] interval, as well as lower than that with Hum milk in the [120′–240′] interval.

The (Phe+Tyr) iAUC in the [0′–120′] iAUC was not different between the four milk types, whereas in the [120′–240′] interval the iAUC with Hum milk was significantly lower than that with Cow milk (*p* < 0.003), and nearly significantly lower than that Cow [↓Cas] milk (*p* = 0.054) (Figure 6c).

There was no significant difference between the four groups in the NEAA iAUC in either the [0′–120′] or the [120′–240′] time intervals (Figure 6d).

No significant correlation was found between the increase of plasma amino acid concentrations (both as total and as the BCAA only) and the increase of either insulin, C-peptide or incretin concentrations within the selected time intervals (i.e., in the 5′–20′, 10′–30′, and 60′–120′ integrated areas under the curve, iAUC).

## 4. Discussion

In this study, we show that the early insulin and C-peptide responses to the iso-lactose loads of natural human and cow milk in young healthy volunteers are similar, despite the much lower protein content of human milk. This observation is relevant because the early (i.e., within the first ≈10′–30′) insulin secretion represents the key homeostatic factor to controlling postprandial plasma glucose concentration following nutrient ingestion [33]. Notably, both glucose and amino acids synergistically cooperate to stimulate the early insulin secretion [19,21]. Such a specific potency of human milk appeared somehow intriguing and deserved further investigation. Our study, while confirming the seminal observation by Gunnerud et al. [18], was designed to provide additional insight into the possible mechanisms for such a “specific” potency of human milk. We focused our study on the role on incretins and plasma amino acids, by also exploring the effects of casein in experimentally modified human and cow milk, the former casein-added, the latter casein-deprived.

Under the iso-lactose conditions of this study, the response of incretins, i.e., the gastrointestinal hormones that cooperate with glucose and amino acids in the enhancement of insulin secretion [23], do not appear to be responsible for the equal potency of human and cow milk in the insulin stimulation, despite the ≈70% lower protein concentration of the former (Table 1). As a matter of fact, the incretin concentrations in the early (i.e., between [5′–20′]) phase following human milk administration are not greater, rather lower, than those following cow milk, and no greater either, thereafter (Figure 3 and Figure 4). In the referenced study [18], the incretin data were incompletely reported, therefore preventing a full comparison between human and cow milk. Secondly, we show that the experimental increase of casein (henceforth, of total protein) concentration in human milk, to a value close to that of natural cow milk, does not impair its capacity to enhance the insulin and incretin responses. This finding excludes a possible interference by casein itself, added to human milk, on the insulin-stimulatory effect conveyed by whey proteins and/or glucose. Conversely, the experimental reduction of casein in natural cow milk, to a value close to that of human milk, impairs its early insulin and incretin responses. Third, neither plasma amino acids are apparently involved in the equally potent insulin and C-peptide stimulation of human and cow milk, since plasma aminoacidemia following human milk is not greater, but rather lower, than that following cow milk. Our study, while confirming the seminal results of Gunnerud et al. [18], adds new information useful in the understanding of the possible mechanisms explaining the specific potency of human milk.

Casein deprivation in cow milk reduced the early insulin and C-peptide response (by ≈50% at the 30′ peak), in respect to that of both cow and human natural milks (Figure 2b,c), and despite post-milk plasma amino acid increments similar or even slightly greater following the low-protein cow than human natural milk (Figure 5 and Figure 6). Thus, plasma amino acids were not apparently involved in the reduced insulin early response to the low-protein cow milk administration. However, the lower early [5′–20′] GLP-1 and GIP responses following the low-protein rather than the natural cow milk (Figure 4a,b) suggest a role of casein in sustaining the response of incretins, and possibly, also of insulin, to natural cow milk.

Despite a more marked postingestion hyperaminoacidemia (either as total amino acids or of the BCAA only), within the [5′–90′] interval following the casein-added human milk, the insulin response was not altered, i.e., it was neither increased nor decreased in respect to that of natural human milk. The interpretation of such a finding is difficult and possibly double-faceted. On the one side, while the more marked hyperaminoacidemia after the casein-added human milk should have stimulated insulin secretion further, such a potential, additional effect could have been blunted by some interference at the intestinal level, between the casein itself and the WP-mediated stimulation of insulin secretion (Figure 5b). It is well established that WP is more potent than casein (or cheese) on insulin secretion [21,34], since it behaves as a “fast” protein, more rapidly digested and absorbed than the “slow” protein casein, and leading to a brisker and more marked postingestion hyperaminoacidemia [34,35,36]. It is then possible that the main actor in the stimulation of insulin secretion is the concentration of the whey proteins themselves. As a matter of fact, from the data of Table 1, the calculated WP total concentrations did not differ much among the four milk types. However, such a conclusion would not apply to the low-protein cow milk, that, despite an estimated WP concentration comparable to that of the other milk types, exhibited a lower early insulin response (Figure 2).

The failure to detect significant correlations between the increase of plasma amino acid concentrations (both as total AA and as the BCAA only) and the increase of either insulin, C-peptide or incretin concentrations at selected time intervals, confirms the complexity of the mechanism(s) controlling hormone response following the ingestion of mixed nutrients. In addition, analytical variation(s) in the amino acid analysis, greater than that for hormones (see also Appendix A), could have prevented the detection of significant differences among the experimental groups.

A species-specific effect of the milk proteins on insulin secretion could not be excluded either. In other words, the whey protein fraction of human milk may be more efficient for insulin stimulation in humans, than those from another species, i.e., cow. A species specificity in the protein structure and amino acid sequences does actually exist in human and cow milk proteins [37]. Clearly, this hypothesis requires further studies to be confirmed.

Although casein and whey proteins might modify gastric motility, the existing literature is quite contrasting about their effects [38,39,40]. Unfortunately, in our study we did not measure either gastric motility, curd formation in the stomach, or intestinal amino acid absorption. Nevertheless, amino acid absorption apparently was not impaired by either casein removal or addition, as indirectly judged by the measured plasma amino acid increments following the low-casein or the high-casein cow, and human milk loads (Figure 5 and Figure 6).

An objective limitation of our study is that it has been conducted in young adults, not in infants. A study like ours would not, however, be unfeasible in infants, for ethical and practical limitations. Rather, the present study could offer some elements in the interpretation of the mechanisms of action of human and cow milk on insulin secretion.

The experimental manipulation of cow and human milks could have altered the milks’ physiological assembly. Milk structure is very complex, i.e., lipids are emulsified in membrane-coated globules, whereas the proteins are in colloidal dispersions as micelles [41,42,43]. It cannot be excluded that the milk qualitative composition was altered by the dilution/addition procedures here used.

We extended our study to four hours, to be able to detect possible late-occurring differences among the tested milks, whereas in previous studies the duration was limited to 2 h [18,21]. The longer duration of our study allowed us to better describe the time-dependent pattern of incretin response to milk ingestion. GIP is produced by K-cells of the proximal part of the small intestine, whereas GLP-1 is produced by L-cells located in more distal intestinal tracts [23]. The GIP response to milk ingestion was more clearly biphasic than that of GLP-1 (Figure 3), and may reflects the intestinal location of the incretin-producing cells, as well as the different rates of casein and whey protein digestion.

The lysozyme content of human milk is ≈0.8–1 g/L, accounting for ≈8% of total whey proteins [8], whereas lysozyme is almost undetectable (≈1.5 μg/L) in cow milk [44]. Although it cannot be excluded that the higher lysozyme concentration of natural human rather than cow milk played a specific role in insulin and C-peptide stimulation, to our knowledge no experimental data exist on the effect of lysozyme on insulin secretion in vivo. This hypothesis should, however, be directly tested.

Human milk contains ≈20–25% of total nitrogen-containing substances as non-protein nitrogen [7,8]. In our study, we did not consider nor measure this fraction. The possible role of nonprotein nitrogen on hormone secretion might worth exploring. We also did not measure nor consider the oligosaccharides of milk, particularly those of human milk [7]. Their possible effects on either the glycemic responses or on any interference on substrate intestinal absorption would require specific studies.

The fat content of cow milk was apparently greater than that of natural human milk, of Cow [↓Cas] milk, and of the Hum [↑Cas] milk (Table 1). Whether fat per se stimulates insulin secretion in humans is controversial [45,46]. Nevertheless, one target of the study was to test the cow and human milks head-to-head, i.e., as they were, without any manipulation except for the lactose content. Although we acknowledge that the difference in fat content between these two milk types could theoretically have affected the results, the insulin and C peptide responses were not different between these two milk types, nor between the natural Cow and the Hum [↑Cas] milk. If anything, the greater fat content of cow milk should have resulted in a greater insulin and C-peptide response than that observed with both human milk-containing oral loads, which was not the case. Conversely, there was no difference in fat content between the two low-protein milks (i.e., the Hum and the Cow [↓Cas] milks), another head-to-head comparison of our study.

Oral fat (particularly monounsaturated fatty acids) can stimulate incretin secretion [47,48,49,50]. Therefore, the greater GLP-1 and GIP responses observed at some time points following the cow rather than the other milk types (Figure 3a,b) could well be due to its greater fat content, however with apparently little or no impact on insulin secretion, as discussed above.

## 5. Conclusions

Our study provides novel data on the characterization of the effects of human milk, compared to those of cow milk, on insulin and incretin responses, as well as on the possible role of milk proteins and plasma amino acids on these effects. This information may be helpful to better understand the physiology of hormone secretion following milk ingestion in humans.

## Figures and Tables

**Figure 1 nutrients-14-01624-f001:**
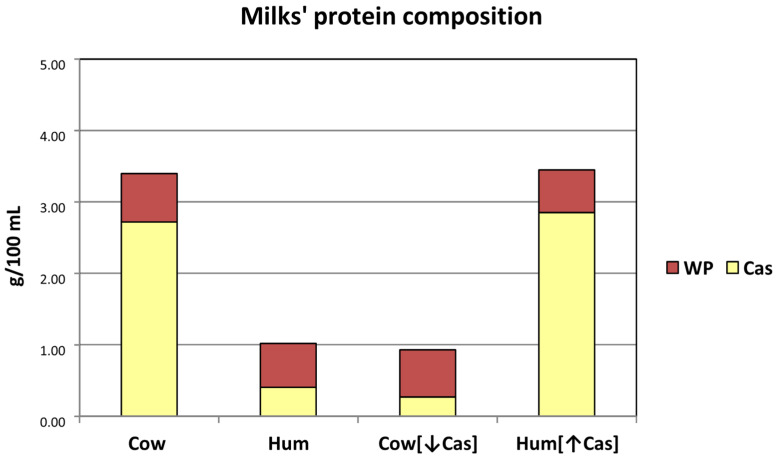
Schematic protein composition of the milk types employed in the study. The Y axis reports the protein concentrations (g/100 mL) of each milk type. The (approximate) casein (Cas) and whey protein (WP) concentrations are reported as either yellow (lower) or brown (upper) part of the main bars. Abbreviations: Cas: casein; Hum: human; Cow [↓Cas]: low-casein cow milk; Hum [↑Cas]: high-casein human milk; WP: whey protein; WP/Cas: whey protein to casein ratio.

**Figure 2 nutrients-14-01624-f002:**
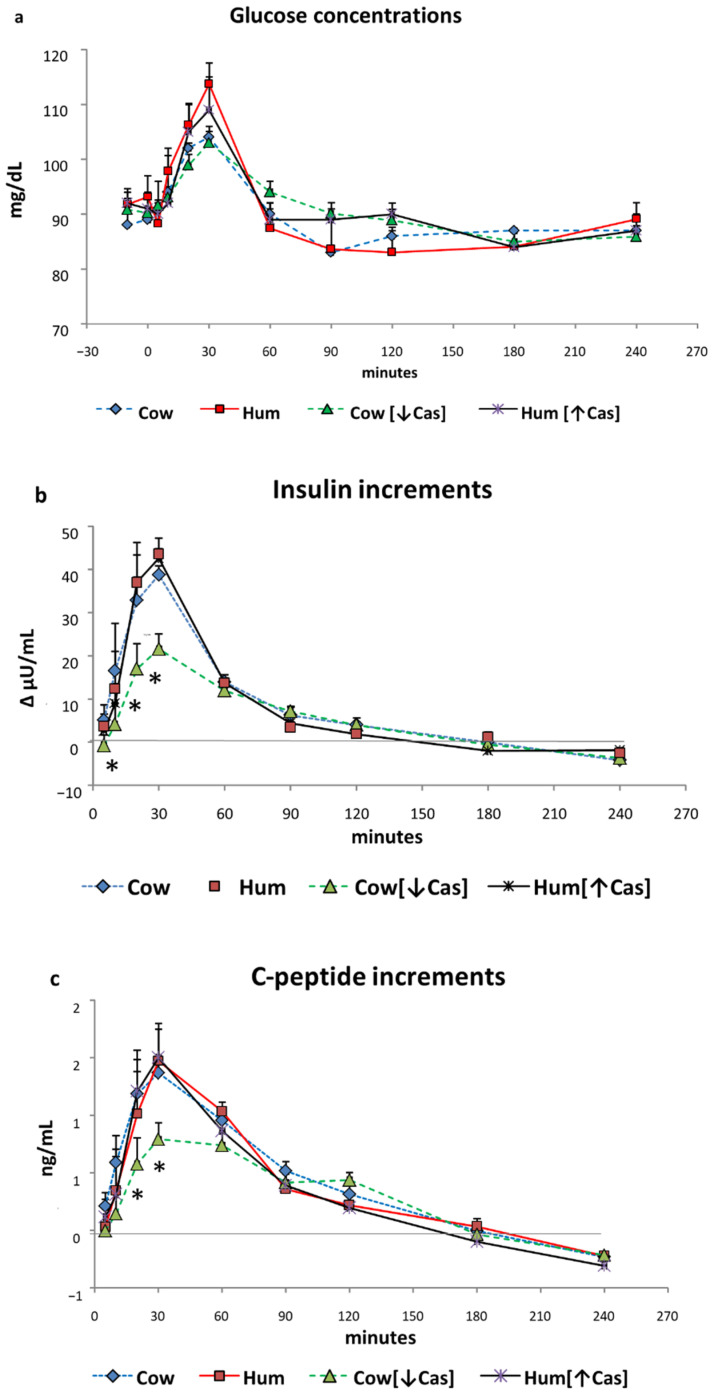
Glucose, insulin, and C-peptide. Abbreviations: Hum: human; Cow [↓Cas]: low-casein cow milk; Hum [↑Cas]: high-casein human milk; WP: whey protein. (**a**): Plasma glucose concentrations (expressed as mg/dL) before and following milk administration over the 240 min study period. Data are mean ± SEM. No significant differences between the four milk types were detected at any time points. Sample size: Cow, *n* = 7; Hum, *n* = 8; Cow [↓Cas], *n* = 10; and Hum [↑Cas], *n* = 7. (**b**): Increments vs. basal values of plasma insulin concentrations (expressed as Δ μU/mL) following milk administration over the 240 min study period. Data are reported as mean ± SEM. The symbol (*) indicates the level(s) of the statistical differences (by 2-way ANOVA and the Fisher’s LSD test) between the Cow [↓Cas] milk and Cow milk at 10′ (*p* < 0.04), as well as between the Cow [↓Cas] milk and either Cow (*p* < 0.006 and *p* < 0.015), or the Hum (*p* < 0.05 and *p* < 0.025) milk at 20′ and 30′ respectively. Sample size: Cow, *n* = 7; Hum, *n* = 8; Cow [↓Cas], *n* = 10; and Hum [↑Cas], *n* = 7. (**c**): Increments vs. basal values of plasma C-peptide concentrations (expressed as Δ ng/mL) following milk administration over the 240 min study period. Data are reported as mean ± SEM. The symbol (*) indicates the level(s) of the statistical difference (by two-way ANOVA and the Fisher’s LSD test) at 20′ and 30′ between the Cow [↓Cas] milk and either the Cow (*p* < 0.012 and *p* < 0.016, respectively), the Hum (*p* = 0.067 and *p* < 0.005, respectively) or the Hum [↑Cas] milk (*p* < 0.01 and *p* < 0.004, respectively). Sample size: Cow, *n* = 7; Hum, *n* = 8; Cow [↓Cas], *n* = 10; and Hum [↑Cas], *n* = 7.

**Figure 3 nutrients-14-01624-f003:**
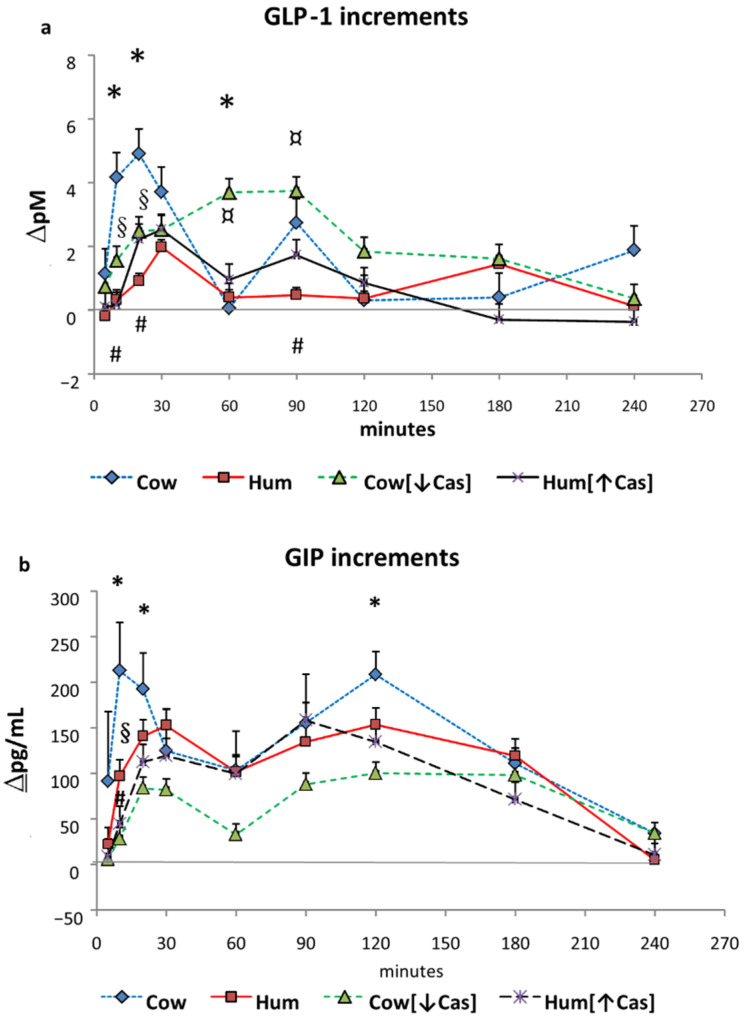
GLP-1 and GIP. Abbreviations: GIP: glucose-inhibitory-polypeptide; GLP-1: glucagon-like polypeptide-1; Hum: human; Cow [↓Cas]: low-casein cow milk; Hum [↑Cas]: high-casein human milk. (**a**): Increments vs. baseline values of plasma GLP-1 concentrations (expressed as Δ pM/mL) following milk administration over the 240 min study period. Data are means ± SEM. The symbols indicate the statistical difference (by two-way ANOVA the Fisher’s LSD test): (*) at 10′, 20′ and at 60′ between Cow [*n* = 7] and Cow [↓Cas] milk (*n* = 9) (*p* = 0.025, *p* < 0.04, and *p* < 0.002, respectively); [#] at 10′, 20′ and 90′ between Cow [*n* = 7], and Hum [*n* = 7] milk (*p* = 0.002, *p* < 0.0015, *p* = 0.066); [§] at 10′ and 20′ between Cow [*n* = 7] and Hum [↑Cas] [*n* = 6] milk (*p* < 0.002 and *p* < 0.04, respectively); and finally [¤] between Hum [*n* = 7] and Cow [↓Cas] milk [*n* = 9] at 60′ and 90′ (*p* < 0.005 at both time points) (by two-way ANOVA the Fisher’s LSD test). (**b**): Increments vs. baseline values of plasma GIP concentrations (expressed as Δ pM/mL) following milk administration over the 240 min study period. Data are means ± SEM. The symbols indicate the levels of statistical difference (by two-way ANOVA the Fisher’s LSD test): (*) at 10′, 20′ and 120′ between Cow [*n* = 7] and the Cow [↓Cas] milk [*n* = 9] (*p* < 0.0001, *p* = 0.015 and *p* < 0.02 respectively); at 10′ between the Cow [*n* = 7] and either [§] the Hum [*n* = 7] (*p* < 0.02) or the [#] Hum [↑Cas] [*n* = 6] (*p* = 0.001) milk.

**Figure 4 nutrients-14-01624-f004:**
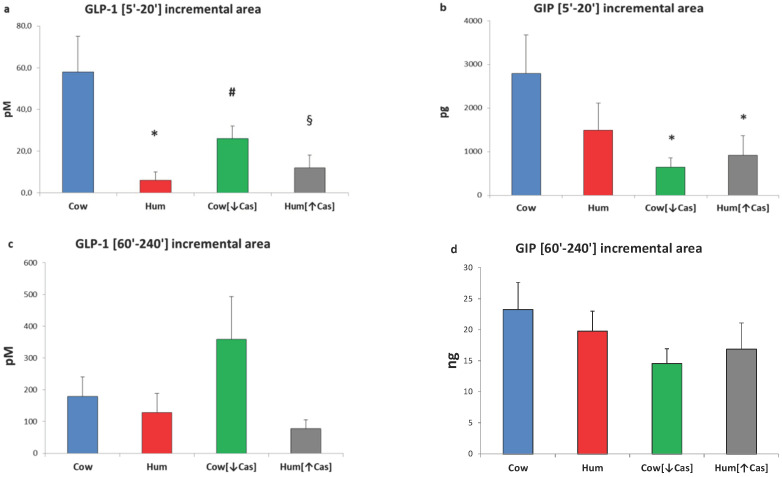
GLP-1 and GIP incremental areas. Abbreviations: GIP: glucose-inhibitory-polypeptide; GLP-1: glucagon-like polypeptide-1; Hum: human; Cow [↓Cas]: low-casein cow milk; Hum [↑Cas]: high-casein human milk. (**a**): Integrated increments of GLP-1 vs. baseline over the [5′–20′] period. The symbols indicate significant differences between Cow milk [*n* = 7] and either [*] Hum (*n* = 7) (*p* < 0.002), [#] Cow [↓Cas] [*n* = 9] (*p* < 0.035) or [§] Hum [↑Cas] milk [*n* = 6] (*p* < 0.007) (by one-way ANOVA and the Fisher’s LSD test). (**b**): Integrated increments of GIP vs. baseline over the [5′–20′] period. The symbol (*) indicates a significant difference between Cow [*n* = 7] and either the Cow [↓Cas] milk [*n* = 9] (*p* < 0.012) or the Hum [↑Cas] milk [*n* = 6] (*p* = 0.03) (by one-way ANOVA and the Fisher’s LSD test). (**c**): Integrated increments of GLP-1 vs. baseline over the [60′–240′] period. There was no significant difference among the four milk types. (**d**): Integrated increments of GIP vs. baseline over the [60′–240′] period. There was no significant difference among the four milk types.

**Figure 5 nutrients-14-01624-f005:**
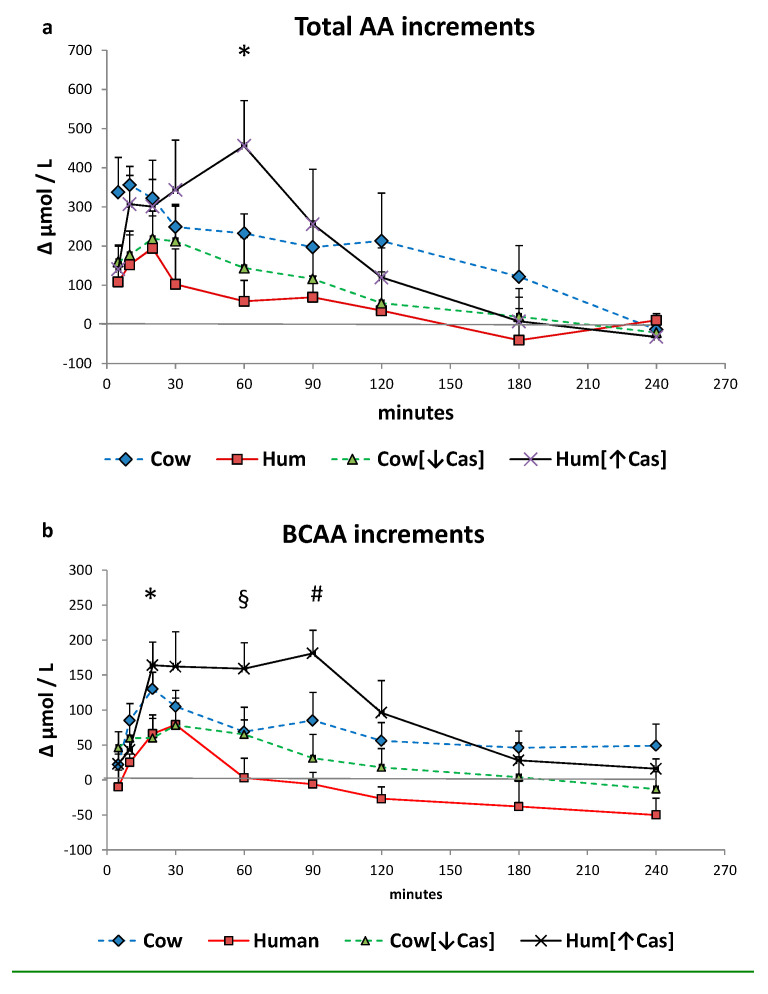
Plasma amino acid increments. Abbreviations: AA: amino acids; BCAA: branched-chain AA; Phe: phenylalanine; Tyr: tyrosine; Hum: human; Cow [↓Cas]: low-casein cow milk; Hum [↑Cas]: high-casein human milk. (**a**): Increments vs. baseline of the sum of total amino acid [AA] concentration over the 4-h study period following milk ingestion. The symbol (*) at 5′ indicates a significant difference (by the two-way ANOVA and the Fisher’s LSD test) between Cow milk [*n* = 7] and either Hum [*n* = 7] [*p* < 0.02], Cow [↓Cas] [*n* = 10] (*p* < 0.025) or Hum [↑Cas] [*n* = 7] (*p* < 0.04) milk. The symbol (#) at 10′ indicates a significant different between Cow and either Hum (*p* < 0.025) or the Cow [↓Cas] (*p* < 0.03) milk (by one-way ANOVA and the Fisher’s LSD test). The symbol (§) at 60′ indicates a significant difference between the Hum [↑Cas] milk, and either Hum (*p* = 0.0025) or Cow [↓Cas] milk (*p* = 0.008), whereas that with Cow milk was of borderline significance (*p* = 0.075) (by one-way ANOVA and the Fisher’s LSD test). (**b**): Increments vs. baseline of the sum of branched-chain amino acids (BCAA) concentration over the 4 h study period following milk ingestion. The symbol (*) at 20′ indicates a significant difference (by one-way ANOVA and the Fisher’s LSD test) between the Hum [↑Cas] [*n* = 7] milk and either Hum [*n* = 7] (*p* < 0.035) or Cow [↓Cas] [*n* = 10] (*p* < 0.016) milks. The symbol (#) at 60′ indicates a significant difference between the Hum [↑Cas] milk and either Hum [*p* < 0.005] or Cow [↓Cas] (*p* = 0.05) milks. The symbol (§) at 90′ indicates a significant difference between the Hum [↑Cas] milk and either Hum (*p* < 0.001) or Cow [↓Cas] [*p* < 0.003] milk (by one-way ANOVA and the Fisher’s LSD test). (**c**): Sum of concentration increments vs. baseline of phenylalanine [Phe] and tyrosine [Tyr] over the 4 hours after milk ingestion. The symbol [*] at 90’ indicates a significant difference between the Hum [↑Cas] [*n* = 7] milk and either the Cow [↓Cas] [*n* = 10] (*p* < 0.001) or the Cow [*n* = 7] milk (*p* < 0.02). The symbol [#] at 180’ indicates a significant difference between Hum milk [*n* = 7] and either the Cow [=7] (*p* < 0.001), the Cow [↓Cas] [*n* = 10] (*p* < 0.015) or the Hum [↑Cas] milk (*p* = 0.02). (**d**): Sum of concentration increments vs. baseline of non-essential amino acids [NEAA] (alanine, glycine, proline, OH-proline, serine, cysteine, glutamic acid, tyrosine) over the 4 h after milk ingestion. There was no significant difference between the groups at any time point.

**Figure 6 nutrients-14-01624-f006:**
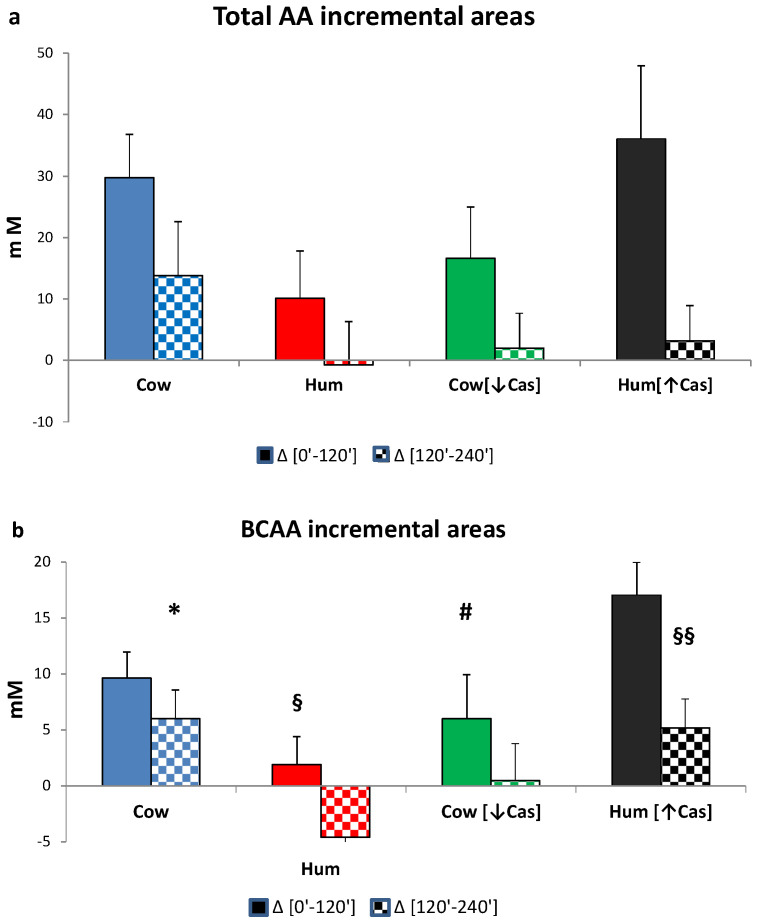
Incremental areas of amino acids. Abbreviations: AA: amino acids; BCAA: branchedchain AA; Phe: phenylalanine; Tyr: tyrosine; Hum: human; Cow [↓Cas]: low-casein cow milk; Hum [↑Cas]: high-casein human milk. (**a**): Integrated incremental areas under the curve (iAUC) over baseline, of total AA concentrations, in the [0′–120′] (left full bars) and the [120′–240′] iAUC (right checkered bars) time intervals, respectively. There was no difference, among the four milk types, in either the [0′–120′] or the [120′–240′] iAUC. The [120′–240′] areas were lower (*p* < 0.0001) than the [0′–120′] ones (by the paired t test) in all milk groups. (**b**): Integrated incremental areas under the curve (iAUC) over baseline, of the BCAA concentrations, in the [0′–120′] (left full bars) and the [120′–240′] iAUC (right checkered bars) time intervals, respectively. The symbol (*) indicates a significant difference (by one-way ANOVA and the Fisher’s LSD test) between Cow and Hum milk (*p* < 0.03) in the [120′–240′] iAUC. The symbol (§) indicates a significant difference (*p* < 0.03) between Hum and Hum [↑Cas] in the [0′–120′] iAUC. The symbol (#) indicates a significant difference (*p* < 0.02) Cow [↓Cas] and Hum [↑Cas] in the [0′–120′] iAUC. The symbol (§§) indicates a significant difference (*p* < 0.04) between Hum and Hum [↑Cas] milk the [120′–240′] iAUC. The [120′–240′] areas were significantly lower (*p* < 0.0001) than the [0′–120′] ones (by paired *t* test). (**c**): Integrated incremental areas over baseline, of the sum of phenylalanine (Phe) and tyrosine (Tyr) concentrations, in the [0′–120′] (left bars, full colors) and the [120′–240′] (right bars, checkered-pattern colors) time intervals, respectively. The symbol (*) indicates a significant difference between Hum and Cow milk in the [120′–240′] (*p* < 0.003), as well as borderline-significant difference between the Hum and the Cow [-Cas] milk (*p* = 0.057). The [120′–240′] areas were significantly lower [*p* < 0.0001] than the [0′–120′] ones (by paired *t* test). (**d**): Integrated incremental areas over baseline, of the sum of non-essential amino acids (NEAA) (alanine, glycine, proline, serine, hydroxyproline, cysteine, glutamate, tyrosine), in the [0′–120′] (left bars, full colors) and the [120′–240′] (right bars, checkered-pattern colors) time intervals, respectively. There were no differences in the iAUC in any study group. The [120′–240′] areas were lower (*p* = 0.001) than [0′–120′] ones (by the paired *t* test) in each experimental group except for the Cow milk group.

**Table 1 nutrients-14-01624-t001:** Volumes and composition of the four milk types employed in the study.

Milk Type	Units	Cow	Hum	Cow [↓Cas]	Hum [↑Cas]
Milk volumes	mL/Kg BW				
natural milk		5.47 ± 0.00	4.82 ± 0.07	0.50 ± 0.00	5.02 ± 0.04
added water		0	0	4.32 ± 0.00	0
Total		5.47 ± 0.00	4.82 ± 0.07	4.82 ± 0.00	5.16 ± 0.04
Milk composition					
Lactose	g/dL				
endog.		4.90 ± 0.00	7.12 ± 0.19	0.48 ± 0.00	6.70 ± 0.10
added		1.63 ± 0.00	/	6.51 ± 0.00	/
Total		6.53 ± 0.00	7.12 ± 0.19	6.99 ± 0.00	6.70 ± 0.10
Protein	g/dL	3.4 ^a^	1.02 ± 0.03	0.92 ± 0.05	3.45 ± 0.02
endog. casein		2.72 ^b^	0.41 ^c^	0.27 ± 0.01 ^d^	0.40 ± 0.00 ^c^
added casein					2.45 ± 0.02
Total casein		2.72 ^b^	0.41 ^c^	0.27 ± 0.01 ^d^	2.85 ± 0.02 ^c^
endog. WP		0.68 ^b^	0.61 ^c^	0.06 ± 0.00 ^d^	0.08 ± 0.01 ^c^
added WP				0.59 ± 0.00	
Total WP		0.68 ^b^	0.61 ^c^	0.66 ± 0.00	0.60 ± 0.01 ^c^
Fat	g/dL	3.5 ± 0.0	2.30 ± 0.26	2.20 ± 0.11	1.82 ± 0.09
Total energy ^e^	kJ	226	155	149	166

Abbreviations: Hum: human; Cow [↓Cas]: low-casein cow milk; Hum [↑Cas]: high-casein human milk; endog: endogenous (i.e., derived from the cow milk volume used); WP: whey protein. If not specified, the reported data are directly measured on milk samples and/or calculated from added volumes or substances’ weight. ^a^ From the product commercial label protein concentration. ^b^ Calculated from product commercial label protein concentration, and assuming a ratio of (80/20) between casein and total whey protein concentration in cow milk [4]. Such a ratio was very similar to that determined by direct analysis in our laboratory (See Appendix A). ^c^ Calculated from measured total protein concentration in human milk, and assuming an average casein to WP ratio of (≈40/60), of human milk in the first six months of lactation [7]. Such a ratio was very similar to that determined by direct analysis in our laboratory (See Appendix A). ^d^ Calculated from the product commercial label total protein concentration (3.4%, g/dL, w/vol), from the (80/20) ratio between casein and total whey protein in cow milk [4], from the cow milk volume used in the assembly of the Cow [↓Cas] milk, and following the addition of distilled water to dilute and decrease the natural cow milk casein concentration to the desired value. ^e^ assuming 37 kJ/g fat, 13 kJ/g protein, and 8 kJ/g fully-fermentable carbohydrate (Ref. [26]).

## Data Availability

Data of this study can be found upon request to corresponding author.

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
