# Peer review of "Neither Incretin or Amino Acid Responses, nor Casein Content, Account for the Equal Insulin Response Following Iso-Lactose Loads of Natural Human and Cow Milk in Healthy Young Adults"

_nutrients, 2022, doi:10.3390/nu14081624_

Round 1

Reviewer 1 Report

Comments to the authors of the manuscript # nutrients-1635750

The study reported in the manuscript #nutrients-1635750 aimed to decipher why human and cow iso-lactose milks induced the same post-ingestion insulin curves despite lower hyperaminoacidemia after the ingestion of human milk compared to cow milk. The clinical study was registered in ClinicalTrials (NCT04698889), and entitled Effect of Ingestion of Human, Cow, and Modified-cow Milk, on Glucose and Hormone Responses in Humans (PROLAT). The primary outcomes mentioned were plasma insulin and c-peptide, and secondary outcomes were plasma incretins (glucose-inhibitory polypeptide (GIP) and glucagon-like-polypeptide-1 (GLP-1)). Four milks were compared: natural cow and human milks, casein-deprived cow milk and human milk enriched with cow casein. The authors concluded that “neither casein milk content, nor incretin or amino acid concentrations, account for the specific potency of human milk on insulin secretion”.

Overall the manuscript provides little insight into the role of incretins and amino acids on post-ingestion insulinemia after milk consumption. This reviewer suspects that the data were not obtained in accordance to the journal’s standards because of the following major points.

The clinical study was based on convoluted hypotheses (lines 56 to 68). No references have been cited to support that secretory effects of glucose and amino acids are additive, on which the protocol seems to be based. Additionally, it appears that the study implementation deviates from what is described in ClinicalTrials. The test milks published in ClinicalTrials were:

- Human natural milk,

- Cow natural milk,

- A low-protein, cow-derived milk, similar to the human one for total protein content, but with a casein-to-whey protein ratio of casein typical of human milk (20:80);

- A high-protein, cow-derived milk, similar to natural cow milk for total protein content, but with a casein-to-whey protein ratio of casein typical of human milk (20:80).

Could the authors explain why the high protein cow-derived milk to be tested was replaced by a high-casein human-derived milk? Has this change been approved by the Ethical Committee?

The clinical study started in April 2015, was intermittently halted, notably during the covid-19 pandemia, and the whole protocol was not completed as planned. The authors justified a posterior that the number of experiments performed was sufficient according to the statistical power analysis reported in the manuscript (see lines 240-243).

It appears that the composition of the human milk used for the study differ from initial expectations due collections during late-lactation instead of the beginning of lactation (up to 6 months instead of 4 weeks) and the quality controls carried out a posterior.

Concerning the statistical analysis, the global effects of time, milk and the interaction between time and milks (two-way Anova) were not systematically reported. Thus, some data may have been over interpreted when interactions were not significant.

Author Response

Nutrients Editorial Office

28 mar 2022, 10:27 (5 giorni fa)

a me, Alessandro, Monica, Elisabetta, Anna, Emiliano, Elisabetta, Monica, Giovanna, Massimo, Nutrients

Manuscript ID: nutrients-1635750

Reviewer 1

Open Review

English language and style

( ) Extensive editing of English language and style required
( ) Moderate English changes required
( ) English language and style are fine/minor spell check required
(x) I don't feel qualified to judge about the English language and style

Yes

Can be improved

Must be improved

Not applicable

Does the introduction provide sufficient background and include all relevant references?

( )

( )

(x)

( )

Is the research design appropriate?

( )

( )

(x)

( )

Are the methods adequately described?

( )

(x)

( )

( )

Are the results clearly presented?

( )

(x)

( )

( )

Are the conclusions supported by the results?

( )

( )

(x)

( )

Comments to the authors of the manuscript # nutrients-1635750

Reviewer’s Comment:

The study reported in the manuscript #Nutrients-1635750 aimed to decipher why human and cow iso-lactose milks induced the same post-ingestion insulin curves despite lower hyperaminoacidemia after the ingestion of human milk compared to cow milk. The clinical study was registered in ClinicalTrials (NCT04698889), and entitled Effect of Ingestion of Human, Cow, and Modified-cow Milk, on Glucose and Hormone Responses in Humans (PROLAT). The primary outcomes mentioned were plasma insulin and c-peptide, and secondary outcomes were plasma incretins (glucose-inhibitory polypeptide (GIP) and glucagon-like-polypeptide-1 (GLP-1)). Four milks were compared: natural cow and human milks, casein-deprived cow milk and human milk enriched with cow casein. The authors concluded that “neither casein milk content, nor incretin or amino acid concentrations, account for the specific potency of human milk on insulin secretion”.

Overall the manuscript provides little insight into the role of incretins and amino acids on post-ingestion insulinemia after milk consumption. This reviewer suspects that the data were not obtained in accordance to the journal’s standards because of the following major points.

The clinical study was based on convoluted hypotheses (lines 56 to 68). No references have been cited to support that secretory effects of glucose and amino acids are additive on which the protocol seems to be based.

Authors’ Response:

On our opinion, this study adds new data addressing the issue of the human vs. cow milk effect on insulin stimulation in humans. While we may agree about a substantially “negative” conclusion of this study, we anyway exclude both a role of incretins and/or of the amino acids in sustaining insulin secretion following human milk ingestion, and any effect due to its low content in casein. Nevertheless, in order to take into account of such a “negative” result, we would modify the title of the MS, as suggested by one Reviewer, as follows:

“Neither incretin or amino acid responses, nor casein content, account for the equal insulin response following iso-lactose loads of natural human and cow milk in healthy young adults”. 

In regard of the other comment, i.e. that “No references have been cited to support that secretory effects of glucose and amino acids are additive on which the protocol seems to be based”, we didn’t get in depth into this issue (indeed central in the background to our study), because we thought that it was well established (see former Ref. 14 by Floyd et. al, and Ref. 16 by Nilsson et al). The sentence of the Introduction of (former) ln. 51-53: “This finding appeared somehow unexpected, because also amino acids, in addition to glucose-derived lactose, are potent stimulators of insulin secretion (14-16)” should have been self-explanatory. Nevertheless, to sustain this point, we have substituted (former) Ref. 14 with another one from the same author(s) with a more explicative title, as well as we cited that by Nilsson et al. earlier.

Furthermore, we have  modified the previous sentence as follows: “This finding appeared somehow unexpected, because also amino acids are potent stimulators of insulin secretion, in addition to, and synergistically with,  glucose-derived lactose”.

Reviewer’s Comment:

Additionally, it appears that the study implementation deviates from what is described in ClinicalTrials. The test milks published in ClinicalTrials were:

- Human natural milk,

- Cow natural milk,

- A low-protein, cow-derived milk, similar to the human one for total protein content, but with a casein-to-whey protein ratio of casein typical of human milk (20:80);

- A high-protein, cow-derived milk, similar to natural cow milk for total protein content, but with a casein-to-whey protein ratio of casein typical of human milk (20:80).

Could the authors explain why the high protein cow-derived milk to be tested was replaced by a high-casein human-derived milk? Has this change been approved by the Ethical Committee?

Authors’ Response:

As a matter of fact, the cited “high protein cow-derived milk” experimental arm was the object of a specific study of our research group, recently published (Toffolon et al, Mol Nutr Food Res. 2021 Dec;65(24):e2100069. doi: 10.1002/mnfr.202100069. Epub 2021 Nov 15).

The “high-casein human-derived milk” was an additional milk group (arm #5) added to the original project registered in the ClinicalTrials site (please go to the site and verify the addition of: “arm # 5: To study the effect of casein addition to human milk in an additional group of participants recruited under the same eligibility criteria, as well as with the aim to measure the same parameters and outcomes, of the approved trial”). The addition of arm #5 had been approved by the Ethical Committee of the Padova University and City Hospital, as stated in the Method Section (former ln. 85-87) (“The study was approved by the Ethical Committee of the Padova University and City Hospital (N° 2861P, on July 8th, 2013, with a followed-on amendment in April 2021).

Reviewer’s Comment

The clinical study started in April 2015, was intermittently halted, notably during the covid-19 pandemia, and the whole protocol was not completed as planned. The authors justified a posterior that the number of experiments performed was sufficient according to the statistical power analysis reported in the manuscript (see lines 240-243).

Authors’ Response:

We acknowledge that the progression and the completion of the whole protocol was rather complex. Unfortunately, it was planned and started with a crossover design, but then it shifted into a “partially” parallel study protocol. The reason(s) why the protocol couldn’t be completed as originally designed, are reported in the Method section (former ln. 86-93, and footnote #1 on page 5).

Although before the study initiation we performed the statistical power analysis as applicable to a crossover design, in the course of the study we realized that only part of the volunteers would have accepted to complete the planned study number. Therefore, we decided to recalculate the required “n” by applying an independent study group power analysis to account for such a shift in the design.

In two experimental groups (i.e. the Hum and the Cow[↑Cas] milk tests), the finally-enrolled number resulted to be greater (i.e. n=8 and n=11 respectively) than that just required by power analysis, because more subjects unpredictably refused (for a variety of reasons, not associated to the test type) to undertake the remaining planned studies. The investigators then tried to start-over again by enrolling new volunteers accepting to participate into the entire protocol, but also in this case partially unsuccessfully. Conversely, in the remaining two arms (i.e. the Cow and the Hum[↑Cas] milk groups) volunteers’ recruitment was interrupted when the critical “n” (calculated from power analysis of independent groups) was attained, since no volunteer would have accepted to undertake any additional test to complete the desired parallel study.

The following  sentence, explaining such a study shift, has been added in the Statistical section:

“Although the original study design was a cross-over one, it shifted to a partially-parallel one, even to one with independent groups, because of the unavailability of some participants to undertake the planned study number, as reported above. Therefore, we adjusted power analysis to that applicable to four independent groups in the course of the study”.

Reviewer’s Comment

It appears that the composition of the human milk used for the study differ from initial expectations due collections during late-lactation instead of the beginning of lactation (up to 6 months instead of 4 weeks) and the quality controls carried out a posterior.

Authors’ Response:

The Reviewer is correct about this point. The reported, quite ample, lactation period of human milk collection was set between the 15° day of lactation (thus clearly excluding “colostrum” collection) up to six months thereafter. Furthermore, we had to randomly mix different milk batches to attain the milk volume(s) required for each volunteers. Although each milk batch provided to us was originally code-labelled and registered as such in our logs, it was not possible for us to reconstruct the exact milk collection time “a posteriori”, also for privacy issues.  Therefore, we had to refer to such an “ample” collection period as a “conservative” reference time, although some human milk batches might have been predominantly derived from the first months of lactation. While we acknowledge such an experimental limitation, we nevertheless consider it irrelevant, because, from Lonnerdal’s study (J Nutr Biochem 2017) the percent contribution of whey proteins and casein to total milk protein is rather constant (i.e.  ≈60% vs ≈40% respectively) over the 1° and the 6° month of lactation.

Concerning the statistical analysis, the global effects of time, milk and the interaction between time and milks (two-way ANOVA) were not systematically reported. Thus, some data may have been over interpreted when interactions were not significant.

Au response:

We have better detailed the two-way ANOVA for repeated measurements here used, by specifying the statistics of the time-effect. The post-hoc comparison(s) among the groups were performed using the Fisher’s LSD at the reported, specific time points.

Other changes by authors:

  • The footnote on (former) page 14 has been rewritten.
  • The digits of the “p” values of the Results have been extended.
  • The statistical analyses reported in the Result section has been better detailed (in regard of the time effect by ANOVA).
  • The Discussion has been modified and enriched following the Reviewers’ comments and suggestion.
  • Table 1 has been completed with the data of total energy, as requested by one Reviewer.
  • New panels of the Non Essential Amino Acids have been added to Figure 5 and 6, and the legends have been updated.
  • Figure 5b: A panel with the increments of the sum of Non Essential Amino Acids (NEAA) has been added. The statistics have been redone and symbols have been corrected.
  • Figure 6b: A panel with the iAUC of the sum of Non Essential Amino Acids (NEAA) has been added. The statistics have been redone and symbols have been corrected.
  • New references have been added and their sequential number modified accordingly.

Reviewer 2 Report

This research explores the role of incretin and casein on insulin stimulation by human and cow milk in healthy young adults. This is an interesting study that provides novel data on the characterization of the effects of human milk, compared to those of cow milk, on insulin and incretin responses. Therefore, it allow a better understanding of the regulation of hormone secretion following  milk ingestion in humans.

The manuscript is well-structured, comprehensive and clear presented; the methodology and analyses are accurate and correctly conducted.

However, there are some issues that require clarifications:

  • In the section 2.3.3. is reported that the lactose was added to milk with the aim to raise its total concentration to approximately 7g/100ml, in a previous section (2.3.1 line 115) is reported the concentration of 7g/100dl. Please be consistent.
  • The differences in glucose concentrations, insulin and C-peptide increments at 30’ and 60’ should be highlighted and discussed in more detail.
  • Several researches have explored, in an animal model, the metabolic regulatory processes underlying the functional differences observed following administration of milk derived from different species. In particular, the studies focused on the modulatory mechanisms involved in glucose and lipid metabolism. These aspects, should be discussed in more detail and appropriately cited.
    doi: 10.3389/fphys.2018.00032 doi: 10.1155/2011/803985 doi: 10.1002/mnfr.201200160 doi: 10.1016/j.jep.2019.112221.
  • The title should be revised.

Minor points

  • Page 4 - Line 173: please check the square bracket.

Author Response

(The authors gave the same response as above.)

Reviewer 3 Report

The manuscript by Tessari et al. on “The role of incretin and casein on insulin stimulation by human and cow milk in healthy young adults” describes in detail an interesting experiment studying the effect of protein in test meals with the same sugar content on the postprandial glucose and insulin levels. Previously observed effects on glucose and insulin were confirmed and it was found that GIP and GLP values seem not of high importance for these findings.

The study is well described and the results contribute to the understanding of the effects of human milk on postprandial metabolite and hormone levels. Nevertheless, there are some questions one could ask.

It seems the study was initially planned with a crossover design, but then was performed as a study with a “partially” parallel study design. How did this affect the statistical power?

How about the energy (fat) content of the various test meals? Were there differences? Could these affect the findings?

Although the volumes consumed seemed similar between the meals, but could even small differences affect gastric emptying?

The test meals were adjusted to body weight; what is the advantage of this compared to a standardized dose, as applied by many other studies? What about differences in body composition (male vs.female, BMI variation)?

Were there any associations/correlation between AUC values of incretins and insulin or amino acids?

Maybe also a comparison of the time courses of essential and non essential amino acids would be of interest?

Specific points

Line 43: what means integrated here?

Line 43: is it really enough to dilute to make cows milk adequate?

Line 82: better in the on April 2015

Line 108: better in than on

Table 1: it seems important to add information on the energy content

Line 173: the casein added to human milk was cows milk casein? Please specify

Line 177: g/kg

Figure 1: I do not understand: “The horizontal stripe on top of the figure reports the WP/Cas ratio in each milk type

Line 224: relative changes: it is not clear what is meant, I assume post meal value – baseline value (difference), but relative could mean post meal value/baseline value (ratio)

Line 242: 0.05%?; 29 studies, maybe better experiments or test meal applications?

Line 297: studies?

Figure 5: graphs might be clearer, if for parameters, which show increments the value 0 for time 0 is included

Line 626: contradicrs with info in line 107, that there were at least three weeks between test meals

Lines 423-430: seems not to agree with Figure 6, maybe text can be shortened to avoid repetition of information in the Figure legend

Figure 1A: right instead of left of the listed proteins?

Lines 840: it would be good to have information on the CV of the analysis of amino acids; now this is somehow mixed with biological variation

Either analytical variation is rather high or biological variation between subjects is for amino acids much higher than for hormones, if I understand correctly

Author Response

(The authors gave the same response as above.)

Round 2

Reviewer 3 Report

I can fully agree with the content of the manuscript as it is now,

thanks for the discussion

There are only typos I would like to mention

line 309: delete directly

line 444: delete the

line 477; of instead of on